# Query K-means Clustering and the Double Dixie Cup Problem

**I (Eli) Chien**
Department ECE
UIUC
ichien3@illinois.edu

**Chao Pan**
Department ECE
UIUC
chaopan2@illinois.edu

**Olgica Milenkovic**
Department ECE
UIUC
milenkov@illinois.edu

## Abstract

We consider the problem of approximate $K$-means clustering with outliers and side information provided by same-cluster queries and possibly noisy answers. Our solution shows that, under some mild assumptions on the smallest cluster size, one can obtain an $(1 + \epsilon)$-approximation for the optimal potential with probability at least $1 - \delta$, where $\epsilon > 0$ and $\delta \in (0, 1)$, using an expected number of $O(\frac{K^3}{\epsilon \delta})$ noiseless same-cluster queries and comparison-based clustering of complexity $O(ndK + \frac{K^3}{\epsilon \delta})$; here, $n$ denotes the number of points and $d$ the dimension of space. Compared to a handful of other known approaches that perform importance sampling to account for small cluster sizes, the proposed query technique reduces the number of queries by a factor of roughly $O(\frac{K^6}{\epsilon^3})$, at the cost of possibly missing very small clusters. We extend this settings to the case where some queries to the oracle produce erroneous information, and where certain points, termed outliers, do not belong to any clusters. Our proof techniques differ from previous methods used for $K$-means clustering analysis, as they rely on estimating the sizes of the clusters and the number of points needed for accurate centroid estimation and subsequent nontrivial generalizations of the double Dixie cup problem. We illustrate the performance of the proposed algorithm both on synthetic and real datasets, including MNIST and CIFAR 10.

## 1 Introduction

K-means clustering is one of the most studied unsupervised learning problems [1, 2, 3] that has a rich application domain spanning areas as diverse as lossy source coding and quantization [4], image segmentation [5] and community detection [3]. The core question in K-means clustering is to find a set of K centroids that minimizes the K-means potential function, equal to the sum of the squared distances of the points from their closest centroids. An optimal set of centroids can be used to partition the points into clusters by simply assigning each point to its closest centroid.

The K-means clustering problem is NP-hard even for the case when K= 2, and when the points lie in a two-dimensional Euclidean space [6]. Moreover, finding an $(1 + \epsilon)$-approximation for $0 < \epsilon < 1$ remains NP-hard, unless further assumptions are made on the point and cluster structures [7, 8]. Among the state-of-the-art K-means approximation methods are the algorithms of Kanungo et al. and Ahmadian et al [9, 10]. There also exist many heuristic algorithms for solving the problem, including Lloyd's algorithm [2] and Hartigan's method [1].

An interesting new direction in K-means clustering was recently initiated by Ashtiani et al [11] who proposed to examine the effects of side-information on the complexity of the K-means algorithm. In their semi-supervised active clustering framework, one is allowed to query an oracle whether two points from the dataset belong to the same optimal cluster or not. The oracle answer to queries

involving any pair of points is assumed to be consistent with a unique optimal solution, and it takes the form "same (cluster)" and "different (cluster)". The method of Ashtiani et al [11] operates on special cluster structures which satisfy the so-called $\gamma$-margin assumption with $\gamma > 1$, which asserts that every point is at least a $\gamma$-factor closer to its corresponding centroid than any other centroid. The oracle queries are noiseless and $O(K \log n + K^2 \frac{\log K + \log(\frac{1}{\delta})}{(\gamma-1)^4})$ same-cluster queries on $n$ points are needed to ensure that with probability at least $1 - \delta$, the obtained partition is the sought optimal solution. Ailon et al. [12] proposed to dispose of the $\gamma$-margin assumption and exact clustering requirements, and addressed the issue of noisy same-cluster queries in the context of the K-means++ algorithm. In their framework, each pairwise query may return the wrong answer with some prescribed probability, but repeated queries on the same pair of points always produce the same answers. Given that no constraints on the cluster sizes and distances of points are made, one is required to perform elaborate nonuniform probabilistic sampling and subsequent selection of points that represent uniform samples in the preselected pool. This two-layer sampling procedure results in a large number of noiseless and noisy queries - in the former case, with running time of the order of $O(\frac{ndK^9}{\epsilon^4})$ - and may hence be impractical whenever the number of clusters is large, the smallest cluster size is bounded away from one, and the queries are costly and available only for a small set of pairs of points. Further extensions of the problem include the work of Gamlath et al. [13] that provides an framework for ensuring small clustering error probabilities via PAC (probably approximately correct) learning, and the weak-oracle analysis of Kim and Ghosh which allows for "do not know" answers [14].

## 1.1 Our Contributions

Unlike other semi-supervised approaches proposed for K-means clustering, we address the problem in the natural setting where the size of the smallest cluster is bounded from below by a small value dependent on the number of clusters K and the approximation constant $\epsilon$, and where the points contain outliers. Hence, we do not require that the clusters satisfy the $\gamma$-margin property, nor do we insist on being able to deal with very small clusters that seldom appear in practice. Outliers are defined as points at "large" distance from all clusters, for which all queries return negative answers and hence add additional uncertainty regarding point placements. In this case, we wish to simultaneously perform approximate clustering and outlier identification. Bounding the smallest cluster size is a prevalent analytical practice in clustering, community detection and learning on graphs [15, 16, 17]. Often, K-means clustering methods are actually constrained to avoid solutions that produce empty or small clusters as these are considered to be artifacts or consequences of poor local minima solutions [18].

Let $\alpha = (\frac{n}{K s_{\min}})$, $1 \leq \alpha \leq \frac{n}{K}$, denote the cluster size imbalance, where $s_{\min}$ equals the size of the smallest cluster in the optimal clustering; when $\alpha = 1$, all clusters are of the same size $\frac{n}{K}$. Furthermore, when the upper bound is met, the size of the smallest cluster equals one.

Our main results are summarized below.

**Theorem 1.1** (Query complexity with noiseless queries). Assume that one is given parameters $\epsilon \in (0, 1)$, $\delta \in (0, 1)$ and $K$, and $n$ points in $\mathbb{R}^d$. Furthermore, assume that the unique optimal clustering has imbalance $\alpha$, where $\alpha \in [1, \frac{n}{K}]$. Then, there exists a same-cluster query algorithm with an expected number of queries $O\left(\frac{\alpha K^3}{\epsilon \delta}\right)$ that with probability at least $1 - \delta$ outputs a set of cluster centers whose corresponding clustering potential function is within a multiplicative factor $(1 + \epsilon)$ of the optimal. The expected running time of the query-based clustering algorithm equals $O(Kdn + \alpha \frac{K^3}{\epsilon \delta})$.

**Theorem 1.2** (Query complexity with noisy queries and outliers). Assume that one is given parameters $\epsilon \in (0, 1)$, $\delta \in (0, 1)$ and $K$, and $n$ points in $\mathbb{R}^d$. Let $p_o$ be the fraction of outliers in the dataset. Furthermore, assume that the unique optimal clustering without outliers has imbalance $\alpha$, where $\alpha \in [1, \frac{n}{K}]$, and that the oracle may return an erroneous answer with probability $p_e < 1/2$. When presented with a query involving at least one outlier point, the oracle always produces the answer "different (cluster)." Then, there exists a noisy same-cluster query algorithm that requires

$$O \left( \frac{\alpha K^4}{\delta \epsilon \, (1 - p_o)(1 - 2p_e)^8} \log^2 \frac{\alpha K^2}{\delta \, (2p_e - 1)^4 \, (1 - p_o)} \right)$$

queries and with probability at least $1 - \delta$ outputs clusters whose corresponding clustering potential function is within $(1 + \epsilon)$ of the optimal. The expected complete running time of the noisy clustering algorithm is bounded from above by $O\left(Kdn + \frac{\alpha K^6}{\delta\epsilon(1-p_o)(2p_e-1)^{10}} \log^3 \frac{\alpha K^2}{\delta\epsilon(2p_e-1)^4(1-p_o)}\right)$, provided that the outliers satisfy a mild separability constraint (see Section 2).

Note that Theorem 1.1 gives performance guarantees in expectation, while Theorem 1.2 provides similar guarantees with high probability. Nevertheless, in the former case, a straightforward application of Markov's inequality and the union bound allow us to also bound, with high probability, the query complexity. In the noiseless setting, we conclude that using $O\left(\frac{\alpha K^3}{\delta\epsilon}\right)$ queries, with probability at least $1 - \delta$ our clustering produces an $(1 + \epsilon)$-approximation. For example, by choosing $\delta = 0.01$, we guarantee that with probability at least $0.99$, the query complexity of our noiseless method equals $O(\frac{\alpha K^3}{\epsilon})$. Compared to the result of Ailon et al. [12], as long as $s_{\min} \geq \frac{n\epsilon^3}{K^7}$, our method is more efficient than the two-level sampling procedure of [12]. The efficiency gap increases as $s_{\min}$ increases. As an illustrative example, let $n = 10^6$, $K = 10$ and $\epsilon = 0.1$. Then, the minimum cluster size constraint only requires the smallest cluster to contain at least one point (since $\frac{n\epsilon^3}{K^7} = 10^{-4} < 1$).

Our proof techniques rely on novel generalizations of the double Dixie coup problem [19, 20]. Similarly to Ailon et al. [12], we make use of Lemma 2 from [21] described in Section 2. But unlike the former approach, which first performs K-means++ sampling and than subsampling that meets the conditions of Lemma 2, we perform a one-pass sampling. Given the smallest cluster size constraint, it is possible to estimate during the query phase the number of points one needs to collect from each cluster so as to ensure an $(1 + \epsilon)$-approximation for all the estimated centroid. With this information at hand, queries are performed until each cluster (representing a coupon type) contains sufficiently many points (coupons). The double Dixie coup problem pertains to the same setting, and asks for the smallest number of coupons one has to purchase in order to collect $s$ complete sets of coupons. The main technical difficulty arises from the fact that the number of coupons required is represented by the expected value of the maximum order statistics of random variables distributed according to the Erlang distribution [20], for which asymptotic analysis is hard when the number of types of coupons is not a constant. In our setting, the number of types depends on $K$, and the number of coupons purchased cannot exceed $n$. To address this issue, we use Poissonization methods [22] and concentration inequalities. Detailed proofs are relegated to the Supplement.

For the case of noisy queries and outliers, our solution consists of two steps. In the first step, we invoke the results of Mazumdar and Saha [23, 24] that describe how to reconstruct all clusters of sufficiently large sizes when using similarity matrices of stochastic block model [25] along with same-cluster queries. The underlying modeling assumption is that every query can be wrong independently from all other queries with probability $p$, and that we cannot repeatedly ask the same query and apply majority voting to decrease the error probability, as each query response is fixed. In the second step, we simply compute the cluster centers via averaging.

In the given context, we only need to retrieve a fraction of the cluster points correctly. Note that the minimum cluster size our algorithm can handle is constrained both in terms of sampling complexity of the double Dixie cup as well as in terms of the cluster sizes that [24] can handle. Additional issues arise when considering outliers, in which case we assume the oracle always returns a negative answer ("different clusters"). Note that if the first point queried is an outlier, the seeding procedure may fail as an answer of the form "different cluster" may cause outliers to be placed into valid clusters. To mitigate this problem, we propose a simple search and comparison scheme which ensures that the first point assigned to any cluster is not an outlier.

We experimentally tested the proposed algorithms on synthetic and real datasets in terms of the approximation accuracy for the potential function, query complexity and the misclassification ratio, equal to the ratio of the number of misclassified data points and the total number of points. Note that misclassification errors arise as the centroids are only estimates of the true centroids, and placements of point according to closest centroids may be wrong. Synthetic datasets are generated via Gaussian mixture models, while the real world datasets pertain to image classification with crowdsourced query answers, including the MNIST [26] and CIFAR-10 [27] datasets. The results show order of magnitude performance improvements compared to other known techniques.

A few comments are at place. The models studied in [11, 24] are related to our work through the use of query models for improving clustering. Nevertheless, Ashtiani et al. [11] only consider ground

truth clusters satisfying the $\gamma$-margin assumption, and K-means clustering with perfect (noiseless) queries. The focus of the work by Mazumdar et al. [24] is on the stochastic block model, and although it allows for noisy queries it does not address the K-means problem directly. The two models most closely related to ours are Ailon et al. [12] and Kim et al. [14]. Ailon et al. [12] focus on developing approximate K-means algorithms with noisy same-cluster queries. The three main differences between this line of work and ours are that we impose mild smallest cluster size constraints which significantly reduce the query complexity both in the noiseless and noisy regime, that we introduce outliers into our analysis, and that our proofs are based on a variation of the double Dixie cup problem rather than standard theoretical computer science analyses that use notions of covered and uncovered clusters. The work of Kim et al. [14] is related to ours only in so far that it allows for query responses of the form "do not know" which can also be used for dealing with outliers.

## 2 Background and Problem Formulation

We start with a formal definition of the K-means problem.

Given a set of $n$ points $\mathcal{X} \subset \mathbb{R}^d$, and a number of clusters K, the K-means problem asks for finding a set of points $\mathbf{C} = \{c_1, ..., c_K\} \subset \mathbb{R}^d$ that minimizes the following objective function

$$\phi(\mathcal{X}; \mathbf{C}) = \sum_{x \in \mathcal{X}} \min_{c \in \mathbf{C}} ||x - c||^2,$$

where $|| \cdot ||$ denotes the $L_2$ norm. Throughout the paper, we assume that the optimal solution is unique, and denote it by $\mathbf{C}^* = \{c_1^*, ..., c_K^*\}$. The set of centroids $\mathbf{C}^*$ induces an optimal partition $\mathcal{X} = \bigcup_{i=1}^{K} \mathcal{C}_i^*$, where $\forall i \in [K], \mathcal{C}_i^* = \{x \in \mathcal{X} : ||x - c_i^*|| \leq ||x - c_j^*|| \; \forall j \neq i\}$. We use $\phi_K^*(\mathcal{X})$ to denote the optimal value of the objective function.

As already stated, the K-means clustering problem is NP-hard, and hard to approximate within a $(1 + \epsilon)$ factor, for $0 < \epsilon < 1$. An important question in the approximate clustering setting was addressed by Inaba et al. [21], who showed how many points from a set have to be sampled uniformly at random to guarantee that for any $\epsilon > 0$ and with high probability, the centroid of the set can be estimated within a multiplicative $(1 + \epsilon)$-term. This result was used by Ailon et.al [12] in the second (sub)sampling procedure. In our work, we make use of the same result in order to determine the smallest number of points (coupons) one needs to collect for each cluster (coupon type). For completeness, the result is stated below.

**Lemma 2.1** (Centroid lemma, Lemma 2 of [21]). Let $\mathcal{A}$ be a set of points obtained by sampling with replacement $m$ points independently from each other, uniformly at random, from a point set $\mathcal{S}$. Then, for any $\delta > 0$, one has

$$P(\phi(\mathcal{S}; c(\mathcal{A})) \leq (1 + \frac{1}{\delta m})\phi^*(\mathcal{S})) \geq 1 - \delta,$$

where $c(\mathcal{A})$ stands for the centroid of $\mathcal{A}$.

In our proof, the Centroid lemma is used in conjunction with a generalization of the double Dixie cup problem to establish the stated query complexity results in the noiseless and noisy setting. The double Dixie cup problem is an extension of the classical coupon collector problem in which the collector is required to collect $m \geq 2$ sets of coupons. While the classical coupon collector problem may be analyzed using elementary probabilistic tools, the double Dixie cup problem solution requires using generating functions and complex analysis techniques. For the most basic incarnation of the problem where each coupon type is equally likely and each coupon needs to be collected at least $m$ times, where $m$ is a constant, Newman and Shepp [19] showed that one needs to purchase an average of $O(K(\log K + (m-1) \log \log K))$ coupons. This setting is inadequate for our analysis, as our coupons represent points from different clusters that have different sizes, and hence give rise to different coupon (cluster point) probabilities. Furthermore, in our analysis we require $m = \frac{K}{\delta\epsilon}$, which scales with $K$ and hence is harder to analyze. The starting point of our generalization of the nonuniform probability double Dixie cup problem is the work of Doumas [20]. We extend the Poissonization argument and perform a careful analysis of the expectation of the maximum order statistics of independent random variables distributed according to the Erlang distribution. All technical details are delegated to the Supplement.

Often, one seeks the K-means solutions in a setting where the cluster points $\mathcal{X}$ satisfy certain separability and cluster size constraints, such as the $\gamma$-margin and the bounded minimum cluster size constraint, respectively. Both are formally defined below.

**Definition 2.2** (The $\gamma$-margin property [11]). Let $\gamma > 1$ be a real number. We say that $\mathcal{X}$ satisfies the $\gamma$-margin property if $\forall\, i \neq j \in [K]$, $x \in \mathcal{C}_i^*$, $y \in \mathcal{C}_j^*$, one has

$$\gamma \|x - c_i^*\| < \|y - c_i^*\|.$$

To describe the cluster size constraint, we now formally introduce the previously mentioned notion of $\alpha$-imbalance.

**Definition 2.3** (The $\alpha$-imbalance property). Let $\alpha \in [1, n/K]$ be a real number. We say that the point set $\mathcal{X}$ satisfies the $\alpha$-imbalance property if $\alpha = \frac{n}{K\,s_{\min}}$.

To avoid complicated and costly two-level queries, we impose an $\alpha$-imbalance constraint on the optimal clustering, excluding outliers.

For the set of outliers, we use a milder version of the $\gamma$-margin constraint, described as follows. Assume that $\mathcal{X} = \mathcal{X}_t \cup \mathcal{X}_o$, where $\mathcal{X}_t$ and $\mathcal{X}_o$ are the nonintersecting sets of true cluster points and outliers, respectively. Outliers are formally defined as follows.

**Definition 2.4.** The set $\mathcal{X}_o$ consists of points that satisfy the $\Gamma(\xi)$-separation property, defined as

$$\forall x \in \mathcal{X}_o,\ \forall\, i \in [K],\ \|x - c_i^*\| > \max_{y \in \mathcal{C}_i^*} \|y - c_i^*\| + \sqrt{\frac{\xi\,\phi^*(\mathcal{C}_i^*)}{|\mathcal{C}_i^*|}} \geq \Gamma(\xi).$$

Here, $\Gamma(\xi)$ stands for the minimum of the lower bounds obtained for all values of $i \in [K]$.

This is a reasonable modeling assumption, as outliers are commonly defined as points that lie in "outlier clusters" that are well-separated from all "regular" clusters. The definition is reminiscent of the $\gamma$-margin assumption, but adapted to outliers. Note that the second term serves as a scaled proxy for the empirical standard deviation of the average distance between cluster points and their centroids. In this extended setting, the objective is to minimize the function $\phi(\mathcal{X}_t, \mathbf{C})$. Furthermore, with a slight abuse of notation, we use $\mathcal{C}_1^*, ..., \mathcal{C}_K^*$ to denote both the optimal partition for $\mathcal{X}_t$ and $\mathcal{X}$. It should be clear from the context which clusters are referred to.

Side information for the K-means problem is provided by a query oracle $\mathcal{O}$ such that

$$\forall x_1, x_2 \in \mathcal{X},\ \mathcal{O}(x_1, x_2) = \begin{cases} 0, & \text{if } \exists i \in [K] \text{ s.t. } x_1 \in \mathcal{C}_i^*, x_2 \in \mathcal{C}_i^*; \\ 1, & \text{otherwise.} \end{cases} \tag{1}$$

Query complexity is measured in terms of the number of times that an algorithm requests access to the oracle. The goal is to devise query algorithms with query complexity as small as possible. The noisy oracle $\mathcal{O}_n$ may be viewed as the response of a binary symmetry channel with parameter $p_e$ to an input produced by a noiseless oracle $\mathcal{O}$. Equivalently, $\forall x_1, x_2 \in \mathcal{X}$, $P(\mathcal{O}_n(x_1, x_2) = \mathcal{O}(x_1, x_2)) = 1 - p_e$, and $P(\mathcal{O}_n(x_1, x_2) \neq \mathcal{O}(x_1, x_2)) = p_e$, independently from other queries. Each pair $(x_1, x_2)$ is queried only once, and the noisy oracle $\mathcal{O}_n$ always produces the same answer for the same query. When presented with at least one outlier point in the pair $(x_1, x_2)$, the noiseless oracle always returns $\mathcal{O}(x_1, x_2) = 1$, while the noisy oracle $\mathcal{O}_n$ may flip the answer with probability $p_e$. The problem of identifying outliers placed in regular clusters is resolved by invoking the algorithm of [24], which places outliers into small clusters that are expurgated from the list of valid clusters.

## 3 Algorithmic Solutions

In what follows, we present two algorithms that describe how to perform noiseless queries and noisy queries with outliers in order to seed the clusters. In the process, we sketch some of the proofs establishing the theoretical performance guarantees of our methods.

The noiseless query K-means algorithm is conceptually simple and it consists of two steps. In the first step, we sample and query pairs of points until we collect at least $\frac{K}{\delta\epsilon}$ points for each of the $K$ clusters. In the second step, we compute the centroids of clusters by using the queried and classified points. The number of points to be collected is dictated by the size of the smallest cluster and the double Dixie cup coupon collector's requirements derived in the Supplement, and summarized below.

---

**Algorithm 1:** Approximate Noiseless Query K-means Clustering

---

**Input:** A set of $n$ points $\mathcal{X}$, number of clusters $K$, an oracle $\mathcal{O}$
**Output :** Estimates of the centers $\mathbf{C}$
Initialization: $t = 1, \mathcal{C}_i = \emptyset, \forall i \in [K], R_i = \emptyset, \forall i \in [K]$.
Uniformly at random sample a point $x$ from $\mathcal{X}, \mathcal{C}_1 \leftarrow \mathcal{C}_1 \cup \{x\}, R_1 \leftarrow x$.
**while** $\min_{i \in [K]} |\mathcal{C}_i| < \frac{K}{\delta \epsilon}$ **do**
    Uniformly at random sample with replacement a point $x$ from $\mathcal{X}$
    **if** $\forall i \in [t], \mathcal{O}(R_i, x) = 0$ **then**
        | $\mathcal{C}_i \leftarrow \mathcal{C}_i \cup \{x\}$
    **else**
        | $t \leftarrow t + 1, \mathcal{C}_t \leftarrow \{x\}, R_t \leftarrow x$
    **end**
**end**
**for** $k = 1$ **to** $K$ **do**
    Let $c_{k,i}$ denote the $i^{th}$ element added to $\mathcal{C}_k, \mu_k = \frac{1}{|\mathcal{C}_k|} \sum_{i=1}^{S_k} c_{k,i}, \mathbf{C} \leftarrow \mathbf{C} \cup \{\mu_k\}$
**end**

---

**Lemma 3.1.** Assume that there are $K$ types of coupons and that the smallest probability of a coupon type $p_{\min}$ is lower bounded by $\frac{1}{\alpha K}$, with $\alpha \in [1, \frac{n}{K}]$. Then, on average, one needs to sample at most

$$2\alpha K (\log K + m \log 2)$$

coupons in order to guarantee the presence of at least $m$ complete sets, where $m = O(K)$.

Note that in our analysis, we require that $m = \frac{K}{\delta \epsilon}$, for some $\epsilon, \delta > 0$, while classical coupon collection and Dixie cup results are restricted to using constant $m$ [20, 19]. In the latter case, the number of samples equals $O(K(\log K + (m-1) \log \log K))$, which significantly differs from our bound.

Two remarks are at place. First, one may modify Algorithm 1 to enforce a stopping criteria for the sampling procedure (see the Supplement). Furthermore, when performing pairwise oracle queries, we assumed that in the worst case, one needs to perform $K$ queries, one for each cluster. Clearly, one may significantly reduce the query complexity by choosing at each query time to first probe the clusters with estimated centroids closest to the queried point. This algorithm is discussed in more detail in the Supplement.

The steps of the algorithm for approximate query-based clustering with noisy responses and outliers are listed in Algorithm 2. The gist of the approach is to assume that outliers create separate clusters that are filtered out using the noisy-query clustering method of [24]. Unfortunately, the aforementioned method assumes that sampling is performed without replacement, which in our setting requires that we modify the Centroid lemma to account for sampling points uniformly at random without replacement. This modification is described in the next lemma.

**Lemma 3.2** (The Modified Centroid Lemma). Let $\mathcal{S}$ be a set of points obtained by sampling $m$ points uniformly at random *without replacement* from a point set $\mathcal{A}$. Then, for any $\delta > 0$, with probability at least $1 - \delta$, one has

$$\phi(\mathcal{A}; c(\mathcal{S})) \leq \left(1 + \frac{1 - \frac{m-1}{|\mathcal{A}|-1}}{\delta m}\right) \phi_1^*(\mathcal{A}) \leq \left(1 + \frac{1}{\delta m}\right) \phi_1^*(\mathcal{A}).$$

Here, $c(\mathcal{S})$ denotes the center of mass center of $\mathcal{S}$, and $m \leq |\mathcal{A}|$.

Furthermore, the requirement that sampling is performed without replacement gives rise to a new version of the double Dixie cup coupon collection paradigm in which one is given only a limited supply of coupons of each type, with the total number of coupons being equal to $n$. As a result, the number of points sampled from each cluster without replacement can be captured by an iid multivariate hypergeometric random vector with parameters $(n, np_1, ..., np_K, m)$. To establish the query complexity results in this case, we do not need to estimate the expected number of points sampled, but need instead to ensure concentration results for hypergeometric random vectors. This is straightforward to accomplish, as it is well known that a hypergeometric random variable may

be written as a sum of independent but nonidentically distributed Bernoulli random variables [28]. Along with tight bounds on the Kulback-Leibler divergence and Hoeffding's inequality [29], this leads to the following bound on the probability of sampling a sufficiently large number of points from the smallest cluster.

**Theorem 3.3.** Without loss of generality, assume that $p_1 \leq p_2 \leq \ldots p_K$, where $p_i \in (0,1)$ for all $i$, and $\sum_i p_i = 1$. Furthermore, assume that during the query procedure, $m$ points from $K$ nonuniformly sized clusters of sizes $(np_1, ..., np_K)$ are sampled uniformly at random, without replacement. Then, the probability that at least $m_o = \frac{m \, p_1}{2}$ points $S$ are sampled from the smallest cluster is bounded as

$$P\{S \geq m_o\} \geq 1 - K \exp\left(-\frac{m_o}{4}\right). \tag{2}$$

---

**Algorithm 2:** Approximate Noisy Query $K$-means Clustering with Outliers

---

**Input:** A set of $n$ points $\mathcal{X}$, the number of clusters $K$, a noisy oracle $\mathcal{O}_n$ with output error probability $p_e$, a precomputed value $M$, and probability $p_o$ of outliers.
**Output :** Centroids set $\mathbf{C}$
**Phase 1:** Seed the clusters by running Algorithm 5 for noisy query-based clustering
Uniformly at random sample $M$ points from $\mathcal{X}$ without replacement. The sampled set equals $\mathcal{A}$.
Run Algorithm 5 (described in the Supplement) on $\mathcal{A}$ to obtain a K-partition of $\mathcal{A} = \bigcup_{i=1}^{K} \mathcal{A}_i$.
**Phase 2:** Estimate the centroids
For all $i \in [K]$, $c_i \leftarrow c(\mathcal{A}_i)$ where $c(\mathcal{A}_i)$ is the center of mass of the set $\mathcal{A}_i$. $\mathbf{C} \leftarrow \{c_1, ..., c_K\}$

---

Recall that the oracle treats outliers as points that do not belong to the optimal clusters, so that in Algorithm 5 described in the Supplement, outliers are treated as singleton clusters. In this case, the minimum cluster size requirement from [24] automatically filters out all outliers. Nevertheless, nontrivial changes compared to the noisy query algorithm derived from [24] are needed, as the presence of outliers changes the effective number of clusters. How to deal with this issue is described in the Supplement.

# 4    Experiments

**Synthetic Data.** For our synthetic data experiments, we start by selecting all relevant problem parameters, the number of clusters K, the cluster imbalance $\alpha$, the dimension of the point dataset $d$, the approximation factor $\epsilon$ and the error tolerance level $\delta$. We uniformly at random sample K cluster centroids in the hypercube $[0,5]^d$ – this choice of the centroids allows one to easily control the overlap between clusters. Then, we generate $n_i$ points for each cluster $i = 1, \ldots, K$, where the values $\{n_i\}_{i=1}^k$ are chosen so as to satisfy the $\alpha$-imbalance property and so that $n_i \in [1000, 6000]$. The points in the cluster indexed by $i$ are obtained by sampling $d$-dimensional vectors from a Gaussian distribution $\mathcal{N}(0, \sigma_i^2 I)$, with $I$ representing the $d \times d$ identity matrix, and adding these Gaussian samples to the corresponding cluster centroid. When generating outliers, we uniformly at random choose a subset of points of size $p_o \times n$, where $n$ is the total number of points to be clustered. Then we adjust the positions of the points to make sure that they satisfy the $\Gamma(2)$-separation property, described in the previous sections. In the noisy oracle setting, we assume that the oracle produces the correct answer with probability $1 - p_e$, for $p_e \in \left(0, \frac{1}{2}\right)$.

We evaluated our algorithms with respect to three performance measures. The first measure is the value of the potential function. As all our algorithms are guaranteed to produce an $(1 + \epsilon)$-approximation for the optimal potential, it is of interest to compare the theoretically guaranteed and actually obtained potential values. The second performance measure is the query complexity, for which we once again have analytic upper bounds. The third performance criteria is the overall misclassification ratio, defined as the fraction of misclassified data points. We also compared our Algorithm 1 with the state-of-the art Algorithm 2 of [12] for the case that there exists one cluster containing one point only. Recall that [12] does not require the smallest cluster size to be bounded away from one, and may in principle operate more efficiently in settings where clusters of smallest possible size (one) exist. As will be seen from our simulation studies, even in this case, our method significantly outperforms [12].

The results of our experiments for the noiseless setting are shown in Figure 1. As may be seen, our analytic approximation results for the potential closely match the results obtained via simulations. In contrast, the actual query complexity is significantly lower in practice than predicted through our analysis, due to the fact that we assumed a worst case scenario for pairwise queries, and set the number of comparisons to K. For the misclassification ratio, we observe that the general trend is as expected – the larger the number of clusters K, the larger the misclassification ratio. Still, the misclassification error in all tested examples did not exceed $2.9\%$. From Figure 1-(d) we can see clearly that our method performs significantly better than Algorithm 2 in [12] even when $\alpha$ is fairly large. We did not compare our noisy query method with outliers with the noisy sampling method of [12] as the latter cannot deal with outliers.

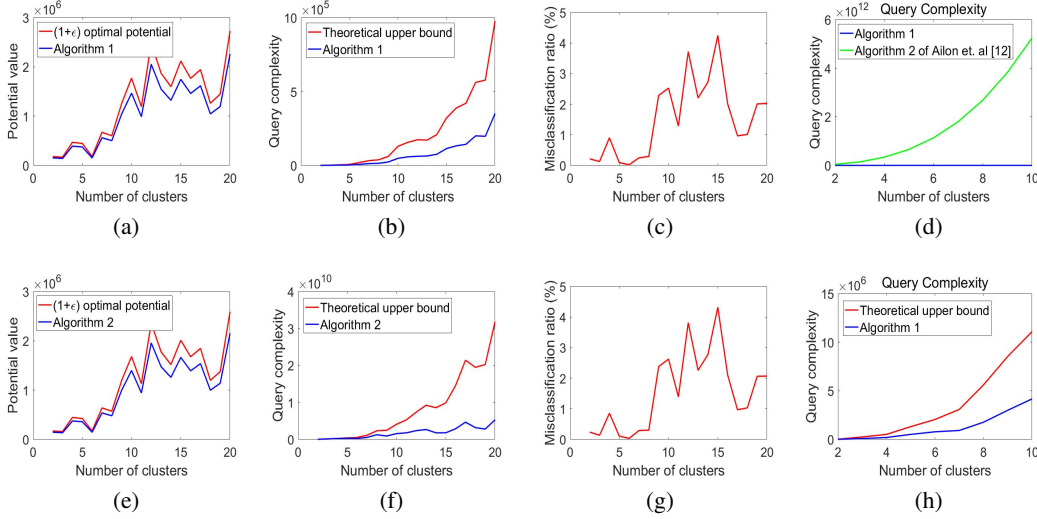

Figure 1: Figures (a) to (c) and (e) to (g) list the results for synthetic data and the noiseless oracle Algorithm 1 and noisy oracle with outliers Algorithm 2, respectively. The parameters are $d = 20, K = [2:20], \alpha = [1,6], \sigma_i = [0,2], \delta = \epsilon = 0.2, p_o = p_e = 0.05$. Figures ((a), (e)) plot the potential, Figures ((b), (f)) the query complexity, and Figures ((c), (g)) the misclassification ratio. Figures (d) and (h) provide comparisons with the noiseless Algorithm 2 of Ailon et. al [12] for a clustering problem with one cluster of size equal to one, with all cluster sizes in the range $[100, 600]$.

Figure 1-(d) reveals that there exists a substantial gap between the query complexity of our method and that of [12] in the noiseless setting. For example, when K=5 and K=10, we require $510, 932$ and $4.16 \times 10^6$ queries. In comparison, Algorithm [12] requires $6.55 \times 10^{11}$ and $5.24 \times 10^{12}$ queries, which in the latter case is roughly a five orders larger number of queries. As a matter of fact, the algorithm in [12] involves a very large constant in its complexity bound, equal to $\frac{2^{23} K^3}{\epsilon^4}$, which for practical clustering settings dominates the complexity expression.

**Real Data.** Since the query complexity of our methods is independent from the size of the dataset, we can provide efficient solutions to large-scale crowdsourcing problems that can be formulated as K-means problems, such as is the case of image classification. We use the following two image classification datasets for which the ground-truth clusters are known and can hence be used to generate the outputs of both the noiseless and noisy oracle:
1) The well-known MNIST dataset [26] comprises $60, 000$ training and $10, 000$ test images of handwritten digits. Each image is normalized to fit into a $28 \times 28$ pixel bounding box and is anti-aliased, which results in grayscale levels.
2) The CIFAR-10 dataset [27] contains $60, 000$ color images with $32 \times 32$ pixels, grouped into $10$ different clusters of equal size, representing $10$ different objects. The clusters are nonintersecting and we sampled $10, 000$ cluster points.
Here, we set $p_o = 0$ and $p_e = 0.05$, hence asserting that there are no outliers, but that $5\%$ of the data points are mislabelled. Note that all the query complexity reported are needed to achieve an $(1 + \epsilon)$-approximation of the potential. The results are shown in Table 1.

Table 1: Real Datasets Results

|  | Actual query complexity | Theoretical query complexity |
|---|---|---|
| MNIST-Algorithm 1 | 12,195 | 38,868 |
| MNIST-Algorithm 2 | 3,628,193,647 | 6,439,271,969 |
| CIFAR 10-Algorithm 1 | 12,490 | 37,479 |
| CIFAR 10-Algorithm 2 | 128,458,964 | 898,432,836 |

## Acknowledgments

This work was supported in part by the grants 239 SBC Purdue 4101-38050 STC Center for Science of Information and NSF CCF 15-27636.

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
