[Reviews · NeurIPS 2018]

Reviewer 1



This paper investigates the problem of active-semi-supervised clustering, by considering both noiseless (perfect oracle) and noisy (imperfect oracle) query responses. The authors provide probabilistic guarantees for low approximation errors to the true optimal k-means objective. The corresponding query complexities are substantially lower than in the existing literature. Importantly, as noted by the authors, their query complexity is independent of the size of the dataset. The main strength of the paper lies in the considerable technical rigour with which the subject has been handled. Numerous results are presented which are either very novel, or, where similar to existing results, provide substantial improvements in terms of the resulting complexities. The experiments while not extensive, are adequate to show considerable improvement over the apparently only available relevant pre-existing method. The paper discusses very technical material, but the authors have done well to elucidate the important points within the text. My major concern lies in the practical relevance of the theory. If I understand correctly the assumption is that the oracle has perfect knowledge of the optimal k-means solution, and not of the true solution. However, this seems quite implausible in most situations. Indeed, as far as I can tell in the experiments the queries are answered based on whether or not the points are in the same true cluster, and not the optimal k-means assignment. Certainly this seems to be a much more realistic implementation, especially in the crowdsourcing scenario mentioned by the authors. That said, the performance on the real datasets is clearly good. It still does not take much of a stretch to imagine very simple examples where the true clusters' means are markedly different from the k-means solutions'. Given the relative infancy of this area, it would be interesting to compare with existing (non active) semi-supervised clustering or semi-supervised classification algorithms (the latter both active and non active). For example, how many true labels are needed for semi-supervised classification to do as well as your approach? This comparison seems relevant if weighing up the relative cost of label queries against pairwise assignment queries. I recorded only one significant typo: in the description of Algorithm 2 there is repetition of "without replacement". ######################################## In the authors' response the justification of the assumptions underlying their theoretical contributions were revisited. Of course I completely recognise that it is necessary to have a well defined mathematical or probabilistic description of the target solution. I also recognise that in many real clustering problems the true cluster labels may not be consistent with any intuitive mathematical formulation of a clustering objective. I would still argue that the assumption of the availability of the optimal k-means solution to an oracle (or crowd) is very unrealistic. I would find it far more satisfying if (even relatively strong) assumptions on the true clusters were made, e.g., that the data arise from a GMM with perhaps an upper bound on the overlap/misclassification rate w.r.t. the true model. Even if this is equally unrealistic in practice, it at least does not create a disconnect between the assumption (that optimal k-means solution is known) and implementation (queries are answered w.r.t. the true clusters and not w.r.t the clusters in the optimal kmeans solution). That said, I am satisfied that this issue is more relevant to the entire literature on this problem and not the paper specifically. Also that there is more than sufficient precedent for these assumptions in high quality publications. I also accept that given the page limit it is not possible to include results comparing active-semi-supervised clustering with semi-supervised classification. The authors' applications in bioinformatics sound extremely interesting and relevant given the importance of same cluster queries being more realistic than label queries. I would find it interesting as a reader to have an example described in the paper where label queries are not relevant. This does not necessitate inclusion of experiments on such a problem, just a description perhaps in the introduction, motivating the relevance of the problem the authors are studying. Given the above, and reading the other reviewers' comments, I more strongly recommend acceptance.

Reviewer 2



There has been a lot of interest recently in clustering using answers to pairwise membership queries of type "Are a and b in the same cluster?" obtained from humans for example via crowdsourcing. This paper considers the problem of finding an (1+\eps)-approximation of k-means using an oracle that can answer such pairwise queries. Both noiseless and noisy oracle settings are studied and upper bounds on number of queries made to obtain an (1+\eps)-approximation to k-means under assumptions on smallest cluster size are provided. Under the noiseless oracle setting, using assumptions on smallest cluster size, it is shown that an (1+\eps)-approx. for k-means can be obtained using O(K^3/\eps) with high probability (where K is the number of clusters). This is an order wise improvement in terms of K and \eps over the previous bound of O(K^9/\eps^4) albeit the previous result doesn't assume any constraints on the cluster sizes. The proof technique used is interesting compared to the existing literature in k-means approximation. Line 49: The order of queries in [12] for noiseless setting is O(K^9/\eps^4). The bound O(ndK^9/\eps^4) is on the run time. For the noisy setting, the algorithm proposed essentially uses the existing work by Mazumdar and Saha[24], to obtain the a clusters (enough points in each clusters) on a subsampled set which is then used to estimate the cluster centers to obtain an (1+\eps)-approximation. The upper bound on the number of queries here seems to be sub-optimal in terms of error probability p_e of the noisy oracle: 1/(1-2p_e)^8 ignoring the log terms. Is this an artefact of using using the algorithm from Mazumdar and Saha[24]? Description of Algorithm 2 is not very clear. Why is A an input while in phase 1 set A is being drawn from X? Naming of phase 1 makes it very confusing as it reads as if Alg 5 is run on A that is input and the whole process is repeated again. Probably, "Seeding" is just enough to describe this phase. While the case with outliers is considered, very stringent constraints are assumed on the outliers as well as the noisy oracle when outliers are involved in the queries. What is the intuition for the second term in Def. 2.4? Line 301-302: Is the comment on the noisy sampling method referring to the noisy oracle? Can you be more specific as to why it substantially differs from your setting? From the description in [12], their noisy oracle (faulty query setting) is exactly same as the one described in this paper (except that [12] does not consider outliers). How do the results in this paper compare with Mazumdar and Saha, NIPS 2017, "Query Complexity of Clustering with Side Information"? Minor comments: ============== 1. Paragraph starting at line 114: There seems to be something missing in the description of this algorithm. It starts off with saying there are two steps and describing the first step. Then the discussion never comes back to say what the second step is. It looks like the second step is essentially computing the centroids on the output of step one, but it is not mentioned. 2. Line 100: The lemma referred to here is stated later Section 2, however it is not mentioned that it is stated later. 3. Line 120: Did you mean to say fraction instead of faction? 4. Line 124: 'queries' should be queried. 5. Line 167: point set S: S here should be script S to be consistent with notation in the equation that follows. 6. Line 200: Def. 2.4: Using \eps here is a bad notation unless this \eps is the same \eps as that used for 1+\eps approximation. 7. Paragraph starting at line 221: Again there is a mention of two steps and the first step is described. The second step is never mentioned. 8. Line 225: Lemma 3.1: Using \alpha \in (0, 1) here is a bad notation. \alpha \in [1, n/K] is very important quantity used to define imbalance of clusters. 9. Line 241-244: Notation for point set A is not consistent. Either use A or script A. ---- I thank the authors for their response to all the reviews. I hope that the description of the algorithm will be revised if the paper is accepted.

Reviewer 3



This paper studies the problem of query k-means clustering, which was initiated in reference [11] (published in NIPS 2016). Query k-means clustering is in a semi-supervised active setting, where queries can be sent to an oracle to receive answers like two points are from the same cluster or different cluster. Following the work in [12] (published in ITCS 2018) which addresses the issue of noisy same-cluster queries in the context of the K-means++ algorithm, this work 1) adds constraints on the smallest cluster sizes to reduce sampling complexity; and 2) identifies outliers that are at large distance from all clusters. Theoretical proofs are provided for the performance guarantees in the case with and without outliers (Theorem 1.1 and 1.2, respectively). The proof is based on generalizations of the double Dixie coup problem. Comparing to [12], the proposed method is more efficient in query complexity. Results in experiments valid the performance of the proposed method on 1) the closeness to the optimal solution, the query complexity and the misclassification rate. The difference to existing work is clearly presented, with theoretical analysis and experimental evaluation. However, the work can be highly improved by including 1) the evaluation of misclassification ratio on real data; The current evaluation on real data is only about the query complexity. It is interesting to know the classification accuracy of the presented method, and that of baseline in [12]. 2) the evaluation of accuracy on outlier identification. Authors claimed that “we wish to simultaneously perform approximate clustering and outlier identification”. Algorithm 5 in the Supplement treats outliers as separated clusters. Can the performance of outlier identification be also evaluated? The paper is generally very well written. There are few typos to correct, like “an framework”, “Theorem 5.3” (should be 1.2). Authors made clarification in the feedback. Wish that the clarification can be included in the revision before publishing.